# Identification of Critical Amino Acids of Coxsackievirus A10 Associated with Cell Tropism and Viral RNA Release during Uncoating

**DOI:** 10.3390/v15102114

**Published:** 2023-10-18

**Authors:** Jie Pei, Rui-Lun Liu, Zhi-Hui Yang, Ya-Xin Du, Sha-Sha Qian, Sheng-Li Meng, Jing Guo, Bo Zhang, Shuo Shen

**Affiliations:** 1Wuhan Institute of Biological Products Co., Ltd., Wuhan 430207, China; pj930608@163.com (J.P.); lrl15549793633@163.com (R.-L.L.); 13552116951@163.com (Z.-H.Y.); duyaxin_duke@163.com (Y.-X.D.); 18868104720@163.com (S.-S.Q.); mengshengli@sinopharm.com (S.-L.M.); guojingwh2004@hotmail.com (J.G.); 2Key Laboratory of Virology, Wuhan Institute of Virology, Chinese Academy of Sciences, Wuhan 430071, China; zhangbo@wh.iov.cn

**Keywords:** coxsackievirus A10, mutants, cell tropism, binding and entry, viral RNA uncoating

## Abstract

Coxsackievirus A10 (CV-A10) is a prevailing causative agent of hand–foot–mouth disease, necessitating the isolation and adaptation of appropriate strains in cells allowed for human vaccine development. In this study, amino acid sequences of CV-A10 strains with different cell tropism on RD and Vero cells were compared. Various amino acids on the structural and non-structural proteins related to cell tropism were identified. The reverse genetic systems of several CV-A10 strains with RD^+^/Vero^−^ and RD^+^/Vero^+^ cell tropism were developed, and a set of CV-A10 recombinants were produced. The binding, entry, uncoating, and proliferation steps in the life cycle of these viruses were evaluated. P1 replacement of CV-A10 strains with different cell tropism revealed the pivotal role of the structural proteins in cell tropism. Further, seven amino acid substitutions in VP2 and VP1 were introduced to further investigate their roles played in cell tropism. These mutations cooperated in the growth of CV-A10 in Vero cells. Particularly, the valine to isoleucine mutation at the position VP1-236 (V1236I) was found to significantly restrict viral uncoating in Vero cells. Co-immunoprecipitation assays showed that the release of viral RNA from the KREMEN1 receptor-binding virions was restricted in r0195-V1236I compared with the parental strain r0195 (a RD^+^/Vero^+^ strain). Overall, this study highlights the dominant effect of structural proteins in CV-A10 adaption in Vero cells and the importance of V1236 in viral uncoating, providing a foundation for the mechanism study of CV-A10 cell tropism, and facilitating the development of vaccine candidates.

## 1. Introduction

Enteroviruses in species A (EV-As) are the primary causes of hand–foot–mouth disease (HFMD), and mutants and recombinants have emerged as a significant public health threat [1]. Recent epidemiological studies indicate that coxsackieviruses A10 and A6 (CV-A10 and CV-A6) are now the dominant pathogens for HFMD, replacing CV-A16 and enterovirus A71 (EV-A71) [2,3]. Both adults and children can be infected by CV-A10, which may cause serious complications such as onychomadesis, hyperkalemia, convulsion, central nervous system disorders, and even death [4,5,6,7,8,9,10]. An epidemiological survey in Xiamen, China, in 2015 also revealed a sharp rise in severe HFMD caused by CV-A10 [9].

Previous studies on enteroviruses such as polioviruses and other HFMD-related enteroviruses indicate that dominant neutralizing epitopes are largely conformational-dependent [11,12]. Inactivated, whole-virion vaccines of polioviruses and EV-A71 vaccines have demonstrated high efficacy in phase III and IV clinical trials and after commercialization [13,14,15]. The efficacies of both inactivated vaccines of polioviruses and EV-A71 are excellent, as humoral immunogenicity is highly related to the conformationally dependent epitopes on whole viral particles. Isolation of virus strains is of the utmost importance for the development of inactivated vaccines. Historically, most coxsackievirus A species such as CV-A2, A4, A5, A6, A8 and A10 were difficult to isolate in cell cultures, though they could infect newborn mice [16,17]. Early studies indicated that cell or tissue tropism of enteroviruses might largely depend upon the presence, absence, abundance, and distribution of receptors. A wide range of cells and culture techniques have been used and developed for the isolation of enteroviruses for diagnosis and laboratory characterization [18]. Both human rhabdomyosarcoma (RD) cells and African green monkey kidney (Vero) cells are continuous cell lines used in HFMD-related virus isolation. RD cells are tumor-derived, and are not approved for the production of human vaccines. After long-term studies on their transformation and tumorigenic potential, Vero cells have been approved and widely used for the production of human vaccines, such as inactivated SARS-CoV-2, poliovirus, and EV-A71 vaccines and live, attenuated rotavirus vaccines [19,20,21].

Compared to EV-A71 and CV-A16, most of the HFMD-related enteroviruses in species A are difficult to isolate in Vero cells, including CV-A10 [2]. Efficient infection with enteroviruses relies on each stage of the virus life cycle [22]. Previous studies have revealed various receptors for different enterovirus species that are used for cell attachment, entry, and uncoating [23,24]. The absence of receptors or low binding efficiency may be vital reasons for the unique difficulty of virus isolation of different enterovirus species. EV-A71 and CV-A16 use scavenger receptor class B member 2 (SCARB2) as the primary receptor, while CV-A6 and CV-A10 use Kringle-containing transmembrane protein 1 (KREMEN1) [25].

CV-A10 is a non-enveloped positive-sense RNA virus belonging to the *Enterovirus* genus in the *Picornaviridae* family [26]. The genomic RNA of CV-A10 encodes a single large polyprotein that self-processes to produce mature viral proteins. The coding region is divided into three parts: P1, P2, and P3. P1 encodes four structural proteins to form the capsid, among which VP1, VP2, and VP3 are exposed to the surface of capsid, while VP4 is an internal protein. The mature virions undergo conformation changes upon receptor binding at low pH, facilitating the release of viral RNA (vRNA) into the cytosol [27]. P2 and P3 encode replication-related, non-structural proteins, including 2A, 2B, 2C, 3A, 3B, 3C and 3D [28,29].

In this study, CV-A10 strains with varying cell tropism were compared to identify the potential key amino acid residues on the structural proteins related to cell adaptation. Using reverse genetics, CV-A10 recombinants with structural protein replacement or site mutations were constructed, and different stages of the virus life cycle were examined to explore the function of the key amino acid residues.

## 2. Materials and Methods

### 2.1. Cells, Viruses and Antibodies

Human rhabdomyosarcoma (RD) cells, African green monkey kidney (Vero) cells, and human embryonic kidney (HEK293) cells were bought from the American type culture collection (ATCC) and maintained in our laboratory. RD cells were maintained in minimum essential medium (MEM). Vero cells and HEK293 cells were maintained in Dulbecco’s modified Eagle’s medium (DMEM). All media were supplemented with 10% newborn bovine serum and cells were maintained in a humidified incubator with 5% CO_2_ at 37 °C.

The CV-A10 strains CV-A10-R2316/XY CHN/2017 (abbreviated as R2316), CV-A10-R3485/XY CHN/2017 (R3485), CV-A10-R3471/XY CHN/2017 (R3471) and CV-A10-R3482/XY CHN/2017 (R3482) were isolated in RD cells from the clinical samples of patients of HFMD in Xiangyang, Hubei, China in 2017 [2]. With successive blind passage in RD and Vero cells, R2316 and R3485 started to induce obvious CPE in Vero cells after 15 generations of passage (abbreviated as R2316V and R3485V). R3471 and R3482 were still not infective to Vero cells. The coxsackievirus A10 strain CV-A10-V0195/PX CHN/2019 (abbreviated as V0195) was isolated directly in Vero cells from a clinical sample of a patient with HFMD in Peixian, Jiangsu, China in 2019 [30].

Anti-CV-A10 rabbit polyclonal antibody was prepared in our laboratory [31]. Alexa Fluor 488-conjugated donkey anti-rabbit IgG (H+L) was purchased from Life Technologies, Carlsbad, CA, USA. Anti-flag tag rabbit polyclonal antibody and 4′,6-diamidino-2-phenylindole (DAPI) were purchased from Beyotime, Shanghai, China. Anti-β-tubulin mouse monoclonal antibody, HRP-conjugated affinipure goat anti-mouse IgG (H+L) and HRP conjugated AffiniPure goat anti-rabbit IgG (H+L) were purchased from Boster, Wuhan, China.

### 2.2. Sequencing of CV-A10 Strains

The genomic RNA of CV-A10 strains was extracted using QIAamp Viral RNA Mini Kit (Qiagen, Dusseldorf, Germany), and the cDNA was synthesized using HiScript II 1st Strand cDNA Synthesis Kit (Vazyme, Nanjing, China) in a total volume of 20 μL with 4 μL of RNA. A polymerase chain reaction (PCR) was conducted using Phanta Master (Vazyme, Nanjing, China) to amplify the viral cDNA followed by sequencing (Sangon, Shanghai, China). The sequence was analyzed using Seqman 7.1 (DNAStar, Madison, WI, USA) software. The amino acid sequences of the viral proteins were compared using Megalign 7.1 (DNAStar, Madison, WI, USA) software.

### 2.3. Construction and Generation of Recombinant CV-A10 Strains

The extracted viral genomic RNA of V0195 and R3482 were amplified by RT-PCR and then cloned to the pBR322 vector named pBR322-r0195 and pBR322-r3482, respectively. The T7 RNA polymerase promoter sequence was added before the 5′-NCR of the viral genome, and the poly A sequence, followed by the HdvRZ sequence (Hepatitis delta virus ribozyme sequence, precise automatic cleavage at the 3′ end of cRNA transcripts), was added after 3′-NCR. The P1 replacement and site mutations were introduced into pBR322-r0195 or pBR322-r3482 via site-directed mutagenesis with PCR and a ClonExpress II One Step Cloning Kit (Vazyme, Nanjing, China). The P1 region of V0195 (GenBank accession No. OR555883) and R3482 (GenBank accession No. OR555883) spans from nt 1 to 2586. The wild-type and recombinant plasmids were co-transfected with pCAGGS-T7 plasmids (expressing the T7 polymerase in eukaryotic cells and maintained in our laboratory) into Vero or RD cells to rescue the corresponding reverse genetic recombinant viruses. After 4 days, the transfected cells were frozen–thawed thrice, and the supernatant was collected after the centrifugation. The viruses were further propagated in RD cells for another two passages. During third passage, viruses were titrated in RD cells, and the resultant mutants were confirmed via sequencing.

### 2.4. Fifty Percent of Cell Culture Infective Dose (CCID_50_) Assay

A monolayer of RD cells in 96-well microplates was inoculated with a serial 10-fold dilution of each virus sample in octuplicate and incubated at 37 °C in the incubator, and the cells were observed daily for CPE up to 7 days post-infection. The CCID_50_ of each virus was calculated using the Reed–Muench method.

### 2.5. Immunofluorescence Assay (IFA)

Vero cells were cultured in 24-well plates overnight and then inoculated with recombinant CV-A10 strains at a multiplicity of infection (m.o.i) of 1. Cells were fixed with 4% paraformaldehyde in PBS after incubating for 24 h. Subsequently, cells were washed with PBS three times and permeated with 0.2% (wt/vol) Triton X-100-PBS solution for 20 min at room temperature. Then, cells were washed and blocked with 3% bovine serum albumin (BSA) in PBS at 37 °C for 30 min. After washing, cells were incubated with rabbit anti-CV-A10 polyclonal antibody for 2 h at 37 °C. Then, cells were washed and incubated with Alexa Fluor 488-conjugated donkey anti-rabbit IgG H+L for 1 h at 37 °C, followed by staining with DAPI in PBS for 5 min to label the nucleus.

### 2.6. Viral Binding and Internalization Assays by qPCR

Twelve-well plates were seeded with 1 × 10^5^ Vero cells per well one day before the assay. Cells were pre-cooled at 4 °C for 1 h and washed with ice-cold PBS before virus inoculation. CV-A10 strains were diluted in ice-cold DMEM and then inoculated at a m.o.i of 10 in a volume of 300 μL per well. The twelve-well plate was transferred to 4 °C for one-hour incubation to allow the viral binding. For virus-binding experiments, cells were washed three times with ice-cold PBS and then the total RNA of infected cells was purified using a FastPure Cell/Tissue Total RNA Isolation Kit (Vazyme, Nanjing, China). For the virus internalization assay, experiments were performed as described previously for WNV, AAV and EMCV [32,33]. In brief, after binding, the virus inoculum was removed, and pre-warmed DMEM was added to the plate. Cells were incubated in a 37 °C incubator for 1 h to allow for virus internalization. Cells were then washed with PBS three times and trypsinized for 3 min at 37 °C to remove the virus still bound to the cell surface. Trypsinized cells were then washed three times with PBS and spun at 500× *g* for 5 min. The total RNA of infected cells was purified using a FastPure Cell/Tissue Total RNA Isolation Kit. Reverse transcription was then conducted with the HiScript II Q RT SuperMix for qPCR (Vazyme, Nanjing, China). SYBR Green-based qPCR was performed using a Pro Universal SYBR qPCR Master Mix (Vazyme, Nanjing, China). The primers designed for CV-A10 viral RNA were designed based on the sequence of the reference CV-A10 genome sequence (GenBank accession number MH118035.1) as follows: vRNA-CV-A10-F: 5′-GACTACTTTGGGTGTCCGTGT-3′, vRNA-CV-A10-R: 5′-AGTCGAGACTTGAGCTCCCAT-3′. The primers for β-actin (Gene ID 60) were designed with PrimerBank (https://pga.mgh.harvard.edu/primerbank/ (accessed on 1 October 2006)) as follows: β-actin-F: 5′-CATGTACGTTGCTATCCAGGC-3′, β-actin-R: 5′-CTCCTTAATGTCACGCACGAT-3′. The content of genomic RNA in the binding/internalization of the virus was calculated relative to the level of β-actin using the formula 2^[Ct(β-actin)−Ct (vRNA)]^. The calculated fold change was converted to delta delta Ct. Viral entry ability was evaluated using the ratio of internalized viruses to bound viruses.

### 2.7. Viral Uncoating Assay

Neutral red (NR)-CV-A10 strains were prepared by infecting RD cells with CV-A10 strains in the presence of 10 μg/mL neutral red in the dark. The NR viruses obtained are easily inactivated by light. NR-CV-A10 strains were used for the uncoating assay [34,35]. Vero cells were inoculated with 100 plaque-forming units (PFU) of neutral red/light-sensitive recombinant CV-A10 strains. After incubation at 37 °C for 1 h, cells were either exposed to light or kept in the dark for 15 min. Viruses that have not completed uncoating will be inactivated after light exposure. The plaque assay was conducted at 5–7 days post-infection. Crystal violet was used to stain the virus plaques, and the number of plaques undergoing light and dark exposure treatments were counted. The viral uncoating ability was assessed by dividing light-exposed PFU by dark-kept PFU.

### 2.8. One-Step Growth Analysis for CV-A10 Strains

Twelve-well plates were seeded with 1 × 10^5^ Vero cells per well one day before the assay. Cells were washed with PBS three times before inoculation. CV-A10 strains were diluted in DMEM and then inoculated at a m.o.i of 10 in the volume of 300 μL per well. After incubation at 37 °C for 1 h, the culture supernatant was removed and the infected cells were washed with PBS thrice before the fresh DMEM was added to each well. The viral culture was frozen–thawed three times at the indicated time points, and the CCID_50_ was calculated based on the Reed–Muench method. Growth curves were generated to compare the propagation characteristics of the CV-A10 strains.

### 2.9. Small Interfering RNA (siRNA) Interference in HEK293 Cells

Three siRNA interfering KREMEN1 were designed and constructed (si-KREMEN1-1 (sense: 5′-CCAUACAAACUUGCAUCAGUUTT-3′, antisense: 5′-AACUGAUGCAAGUUUGUAUGGTT-3′), si-KREMEN1-2 (sense: 5′-GCAGGAUCAUCCUCUUUGAUATT-3′, antisense: 5′-UAUCAAAGAGGAUGAUCCUGCTT-3′) and si-KREMEN1-3 (sense: 5′-GCAUCCAUACAACACUCUGAATT-3’, antisense: 5′-UUCAGAGUGUUGUAUGGAUGCTT-3′)) by Sangon, Shanghai, China. A siRNA-negative control (si-NC) (sense: 5′-UUCUCCGAACGUGUCACGUTT-3′, antisense: 5′-ACGUGACACGUUCGGAGAATT-3′) was established. HEK293 cells were inoculated into a 12-well cell culture plate and transfected with the siRNA when the cell confluence was approximately 80%. After 48 h, the total RNA of the cells of each group was extracted, the mRNA level of KREMEN1 relative to β-actin was detected via qPCR to evaluate the interference efficiency. The HEK293 cells were inoculated with r0195 at a m.o.i of 1 at 48 h post-transfection, and the CPE was observed at 2 days post-infection.

### 2.10. Western Blotting and Co-Immunoprecipitation

The plasmid pcDNA3.1-KREMEN1-flag expressing the KREMEN1 receptor was constructed. HEK293 cells were transfected with pcDNA3.1-KREMEN1-flag using Lipofectamine 2000 (Invitrogen, Carlsbad, CA, USA) according to the manufacturer’s instructions. The viruses were inoculated into cells 2 days post-transfection at a m.o.i of 10. After 1 h of incubation at 37 °C, the cells were washed with PBS and harvested for further analysis as indicated.

For Western blotting, cells were harvested and lysed in RIPA lysis buffer for 30 min on ice and centrifuged at 4 °C at 14,000× *g* for 15 min. Twenty percent of the supernatant was boiled in 1 × loading buffer at 100 °C for 10 min, followed by separation through 4-to-12% gradient sodium dodecyl sulfate-polyacrylamide gel electrophoresis (SDS-PAGE). Then, proteins were transferred onto a nitrocellulose membrane. The membrane was incubated with the primary antibodies against flag-tag and β-tubulin after blocking with 5% BSA in PBST, followed with horseradish peroxidase-conjugated secondary antibodies. The proteins were detected with chemiluminescence. Ten percent of the supernatant was subjected to viral RNA extraction to obtain the viral RNA. The amount of the viral RNA was quantified via RT-qPCR.

Immunoprecipitation was conducted with a flag-tag IP/Co-IP kit (Epizyme, Shanghai, China). The remaining 70% of the supernatant was incubated with anti-flag magnetic beads, followed by washing with lysis buffer three times. The beads were finally resuspended in RIPA lysis buffer and divided into two parts. One portion was subjected to Western blotting to detect IP bands, and all of the other beads were subjected to viral RNA extraction. The amount of the viral RNA was quantified using RT-qPCR.

### 2.11. Statistical Analysis

Graphs were plotted and analyzed using GraphPad Prism 8.0 software (GraphPad, San Diego, CA, USA). Experimental data are presented as the mean ± deviation (SD) for a minimum of three biological replicates. The study utilized a nonparametric one-way ANOVA analysis followed by multiple comparisons of Dunnett’s type to examine the significant differences between each group. Differences were denoted no significance (ns), *p* ≥ 0.05; *, *p* < 0.05; **, *p* < 0.01 and ***, *p* < 0.001.

## 3. Results

### 3.1. Cell Tropism of CV-A10 to Vero Cells

Cell isolation and culture of enteroviruses are pivotal for the basic study and development of inactivated vaccines. The Vero cell is considered non-tumorigenic below a certain passage number, and is safe, recommended and permitted for use as a cell substrate for human vaccines by the WHO and National Regulatory Authorities [21]. RD cells are not approved for the production of human vaccines, for safety reasons. Although RD cells are one of the most susceptible cell lines for human enteroviruses, isolation rates are still very low for *Enterovirus A*, major pathogens for HFMD, except EV-A71 and CV-A16 [36,37]. In our previous study in 2017, a total of 370 CV-A10 viral-RNA-positive clinical specimens were used in cell isolation, only 11 CV-A10 strains were isolated in RD cells, but none of these CV-A10 specimens were isolated in Vero cells [2]. Four RD isolates were then adapted to grow in Vero cells by blind passage. Two of them, R2316V (GenBank accession No. OR555880) and R3485V (GenBank accession No. OR555881) (RD^+^/Vero^+^), were adapted to Vero cells, but the other two, R3471 (GenBank accession No. OR555882) and R3482 (GenBank accession No. OR555883) (RD^+^/Vero^−^), were not (Figure 1a). In another study in 2019, 47 CV-A10-viral-RNA-positive samples were used for cell isolation; one CV-A10 strain, V0195 (GenBank accession No. OR555884) (Vero^+^/RD^+^), was directly isolated in Vero cells from specimens of HFMD patients, which also grew in RD cells (Figure 1a) [30]. The Vero isolate and Vero-adapted strains were also infective to RD cells, and showed extended cell tropism. These studies raise the issues of cell tropism of different genotypes and the adaptation of enteroviruses to different cell lines.

### 3.2. Identification of the Residues Related to CV-A10 Cell Tropism

To investigate the differences in the genomic sequences that determine cell tropism, the complete nucleotide sequences of the five CV-A10 strains, grown or not grown in Vero cells (Figure 1a), were analyzed. We concentrated on variations in the viral proteins though the 5′-noncoding region (NCR); 3′-NCR and nonsense mutations might also play roles in virus propagation in cell culture, and might affect cell tropism [38]. Nucleotide sequence variations in the coding region were identified, and the deduced amino acid sequences of the five strains were aligned and compared (Figure 1b,c). Seven and ten amino acid mutations were located at the structural and nonstructural proteins, respectively. At these positions, Vero-susceptible strains V0195, R2316V, and R3485V harbor the same residues, which are different compared from those of Vero-non-susceptible strains R3471 and R3482. Presumably, these residues may be related to the cell tropism.

### 3.3. Cell Tropism of CV-A10 to Vero Cells Mainly Determined by the Structural Proteins

To confirm the relationship between growth phenotype and genotype, infectious cDNAs were constructed based on the consensus sequences of the Vero strain V0195 and RD strain R3482 stocks (Figure 2a). The parental recombinant r0195 induced CPE, and the infected Vero cells were stained with CV-A10-specific antibodies; however, r3482 did not, consistent with growth phenotypes of the wild-type strains V0195 and R3482. The results confirmed that these variations, identified in consensus sequences of potential quasispecies of wild-type viruses, were closely related to the cell tropism.

To investigate the critical roles played by the variations in structural and nonstructural regions in cell tropism, chimeric P1 replacement recombinants were constructed based on V0195 and R3482 backbones (Figure 2a). All of the four rescued viruses r3482, r0195, r0195-3482P1, and r3482-0195P1 induced obvious CPE, and replicated well in RD cells (Appendix A). The P1 replacement recombinants r0195-3482P1 and r3482-0195P1, as well as parental recombinants r0195 and r3482, were inoculated into Vero cells, and viral protein synthesis was detected via IFA (Figure 2b). Interestingly, it was observed that viral structural proteins were clearly expressed in Vero cells infected with r0195 and r3482-0195P1, while viral proteins were not detected in Vero cells infected with r3482 and r0195-3482P1 (Figure 2b). The results indicated that mutations including the seven amino acid residues in structural protein P1 of V0195, but not those in nonstructural proteins, might play critical roles in growth in Vero cells.

To further explore the mechanism of cell tropism, the binding and internalization of these four recombinants were examined via qPCR assays (Figure 2c,d). When P1 of R3482 replaced the P1 of V0195, the binding of r0195-3482P1 decreased relative to the parental r0195 (*p* < 0.001) (Figure 2c). In contrast, the replacement of the P1 of R3482 with the P1 of V0195 greatly increased the binding of r3482-0195P1 (*p* < 0.001) (Figure 2c). Moreover, the internalization of r3482-0195P1 was dramatically enhanced, while that of r0195-3482P1 was reduced compared with their parental strains r3482 and r0195 (*p* < 0.001), respectively (Figure 2d). The viral entry ability was evaluated with the ratio of internalization to binding (Figure 2e). The entry efficiency of r0195-3482P1 decreased, compared with that of r0195 (*p* < 0.001), to the level of r3482. However, the entry efficiency of r3482-0195P1 increased significantly, compared with r3482 (*p* < 0.01), to the same level of r0195. Viral binding, and entry efficiencies of r0195 or r3482-0195P1 were significantly higher than those of r0195-3482P1 or r3482 in Vero cells (Figure 2c–e), but not in RD cells (except for the enhancement of binding ability of r3482-0195P1 in RD cells) (Appendix A). Both r0195 and r3482-0195P1 could replicate in Vero cells, while r3482 and r0195-3482P1 could not (Figure 2f). The results demonstrate the critical roles in cell tropism played by variations in the P1 structural protein region.

### 3.4. Seven Mutations on the P1 Region Reversing the Cell Tropism

To confirm the role played by the seven mutated residues on the structural proteins in cell tropism, recombinants via site-directed mutagenesis on the P1 region were constructed based on the backbones of r0195 (Figure 3a). The recombinant virus r0195-mut (carrying seven residues of r3482) induced obvious CPE, and replicated well in RD cells, just as the parent strain r0195 did (Appendix A). When compared with the obvious CPE induced by r0195, r0195-mut caused only mild CPE in Vero cells, as only a few cells were rounded up (Figure 3b). As shown in Figure 3c, although some CV-A10-antibody staining spots were also observed in cells infected with r0195-mut, the expression of viral proteins was reduced compared with that of r0195. The viral entry and proliferation ability of r0195-mut were significantly less efficient and lower than those of r0195 in Vero cells (Figure 3d,e). Taken together, these data illustrated that the seven amino acids in the structural proteins were pivotal in the growth of CV-A10 in Vero cells, though other residues in r0195 might help with adaptation too.

### 3.5. V1236 Is Critical for Viral Uncoating in Vero Cells

To investigate the function of each of the amino acid mutations, recombinants with the single-point mutation were constructed, introducing each of the mutations of r3482 into the backbone of r0195 (Figure 4a). Except for mutants r0195-E2143K and r0195-K1141E, the other five recombinants were successfully rescued and replicated well in RD cells, just as the parent strain r0195 did (Appendix A). All rescued recombinants were infectious to Vero cells, but the viral protein staining spots caused by r0195-V1236I were fewer than those caused by the other strains (Figure 4b). The lower expression of VP1 in Vero cells infected with r0195-V1236I, compared to those of the other strains, was also confirmed using Western blotting (Figure 4c). Both r0195-V1236I and r0195-V1283I presented a higher entry efficiency than the parental r0195 (Figure 4d, *p* < 0.01). However, both r0195-V1236I and r0195-R1239K exhibited a lower uncoating ability compared to r0195 (Figure 4e, *p* < 0.001). When the V1236I mutation was introduced, the proliferation of r0195-V1236I was restricted in Vero cells compared with r0195 (Figure 4f). Of all the introduced mutations, r0195-V1236I presented the most significant restriction of viral uncoating and proliferation in Vero cells. Together, these results indicated that a single mutation of the five individual residues did not abolish the infection of Vero cells, but V236 of VP1 was critical for CV-A10 adaption in Vero cells because it affects virus uncoating.

### 3.6. V1236I Mutation Reducing Viral RNA Release during KREMEN1-Dependent Uncoating

Both Vero cells and HEK293 cells are derived from normal kidney tissues. The growth phenotypes of r3482 and r0195 in HEK293 cells were consistent with those in Vero cells; no CPE was observed in r3842-infected and normal HEK293 cells, while typical virus-induced CPE was observed in r0195-infected HEK293 cells (Figure 5a). To investigate the role that the cell receptor KREMEN1 plays in the infection of CV-A10, the KREMEN1 gene was silenced with siRNA in HEK293 cells, which is of higher transfection efficiency than Vero cells. The silencing efficiency of KREMEN1 with the three siRNA could reach 84% (si-KREMEN1-1), 67% (si-KREMEN1-2) and 80% (si-KREMEN1-3) (Figure 5b). With the silencing of the KREMEN1, the CPE in HEK293 cells induced by r0195 was restricted (Figure 5b). KREMEN1 is highly conserved in monkeys and humans. To investigate whether the key role of V1236 in uncoating is related to the cell receptor KREMEN1, the KREMEN1 gene was cloned from Vero cells, and the plasmid pcDNA3.1-KREMEN1-flag expressing the KREMEN1 with a C-terminal flag tag was constructed. The plasmid was transfected to HEK293 cells, followed by inoculation with r3482, r0195 or r0195-V1236I at a m.o.i of 10 for 1 h to complete the viral binding, internalization, and uncoating. Co-immunoprecipitation (Co-IP) experiments were carried out to detect the unreleased viral RNA from the KREMEN1-dependent uncoating. The relative amount of viral RNA bound by KREMEN1 during the uncoating stage was significantly more in r0195-V1236I-infected cells than in r3482 and r0195 (Figure 5c). The results suggested that less viral RNA was released from the KREMEN1-bound virions of the mutant compared to those of the wild-type. It was also noted that over-expression of KREMEN1 in HEK293 cells did not help the infection of r3482 as expected.

## 4. Discussion

Coxsackievirus A10 is one of the major enteroviruses associated with HFMD and threatening the health of infants and young children. It is crucial to uncover the mechanism of cell tropism and adaptation of CV-A10 strains to select viable vaccine candidates in Vero cells for the development of an inactivated vaccine. In our previous epidemiological and etiological research, we isolated only 11 CV-A10 strains out of 370 clinical samples from Xiangyang, China [2]. All the CV-A10 strains exhibited RD cell tropism, not Vero cell tropism, and are unsuitable as vaccine candidates. Fortunately, in our recent study, we obtained Vero-adapted CV-A10 strains via direct isolation or subculture adaptation [30]. By comparing the sequences of the CV-A10 strains with different tropism, we identified certain amino acid residues on the structural and non-structural proteins essential for Vero cell adaptation.

The cell tropism of CV-A10 strains was altered through amino acid substitutions in the P1 region, highlighting the critical function of CV-A10 structural proteins in cell tropism. Structural proteins have multifunctional roles in the enterovirus life cycle [24]. They participate in virion assembly to create an asymmetric icosahedral capsid of enteroviruses. The structural proteins influence viral attachment to the cell receptors and internalization into cells, therefore determining cell tropism [23,39,40]. Virions enter the cytoplasm through receptor-mediated endocytosis, where they undergo an irreversible structural rearrangement to release the viral genome and initiate infection. The uncoating process is also regulated by virion–receptor interactions [41]. The VP1 to VP3 proteins of picornaviruses possess a typical eight-stranded antiparallel β-barrel structure. Mutations in the β-barrels are related to thermostability and viral RNA uncoating [34,42]. In homology remodeling and structural studies of EV-A71, it was reported that the residues on the EF loop of VP2 (residues 136–150) interact with VP1 during scavenger receptor class B2 (SCARB2)-mediated entry to facilitate viral uncoating, impacting EV-A71 fitness in vitro and in vivo [43,44,45]. While SCARB2 is the primary cell receptor for EV-A71, a single-site mutation (L94R on VP1) can modify viral tropism in cell lines and tissues by allowing the virus to bind to the heparan sulfate (HS) attachment receptor [46]. For CV-A10, KREMEN1 is the dominant receptor used for cell attachment, and uncoating and is a rate-limiting factor for infection [25,27]. The structure of CV-A10 and KREMEN1 complexes has been determined, revealing that KREMEN1 binds CV-A10 with a relatively large footprint. The KR domain of KREMEN1 binds the narrow part of the canyon, connecting the C-terminal of VP3, the EF loop of VP2, and the HI loop of VP1. Meanwhile, the WSC domain of KREMEN1 binds the wider region of the canyon, encompassing the BC and GH loops of VP3, as well as the EF loop, GH loop, and C-terminal of VP1 [47]. The KREMEN1 used in our study was cloned from Vero cells, and the amino-acid sequence was aligned with the human KREMEN1. The amino-acid sequence of the KR, WSC and CUB domains of KREMEN1 is highly conserved in monkeys (accession number NP_073246) and humans (accession number NP_114434.3).

The seven amino acids identified in our study are situated at various locations of the structural proteins VP1 and VP2, including the FG loop of VP2, the N-terminal, the DE loop, the HI loop, and the C-terminal of VP1. The N14 and V23 residues are located at the N-terminal of VP1 and are folded within the viral capsid, whereas K141, V236, R239, V283 of VP1, and E143 of VP2 are located at different loops or the C-terminal of the structural proteins, and are exposed on the surface of the viral capsid. Capsid residues VP1 V23 and V283, identified in clade E CV-A10 viruses and implicated in more severe infections, do not directly interact with the receptor, but residue 283 might impact receptor binding by destabilizing the C-terminus of VP1 [9,47]. Cryo-EM observations of different CV-A10 particles have revealed that the N-terminal (residues 1–71) of VP1 lies on the inner surface of the CV-A10 mature virus, while the A-particle, an intermediate of an uncoating particle, transits the N-terminus to exit the capsid [48]. It has been suggested that “umbilical” densities observed in poliovirus are formed by the VP1 N-terminus and VP4, which are involved in delivering the viral genome to the cytoplasm [49]. The r0195-V1236I mutant displays significantly enhanced cell entry ability, and lower uncoating and proliferation efficiencies compared to the parental strain r0195 in Vero cells, and only causes mild CPE. Our findings also reveal that the V1236I mutation of CV-A10 restrains vRNA release from the virions bound to the receptor KREMEN1 during the uncoating phase. We hypothesize that the conformation change of the V1236I mutant, from the virion to the A-particle, was not efficient, and therefore affected the release of viral RNA compared with that of the wild type. Although the introduction of seven site mutations may alter the adaptability of CV-A10 strains to Vero cells, it is unable to completely alter the cell tropism of the CV-A10 strains. The cell tropism of CV-A10 is determined by a variety of factors, including diverse cell receptors and multiple amino acid residues in different viral proteins, making its resolution a more intricate process than anticipated. Considering that some CV-A10 strains are sensitive to RD cells but not Vero cells, it is also speculated that there may be novel CV-A10-related receptors to be discovered in tumor-derived RD cells.

In terms of non-structural proteins, the interactions between the viral RNA and these proteins are crucial for viral replication. Our study identified ten residues in the non-structural proteins that might be related to cell adaption, particularly in the 3C and 3D proteins. The 3C protein is a protease performing viral polyprotein processing by cleaving the peptide bond at Gln-Gly junctions to produce individual virus proteins [50,51]. Additionally, the 3C protein binds to various regions of the viral RNA, including the 5′-NCR, 3′-NCR, and the coding region of the 2C protein [52,53,54]. The binding of 3C to the stem-loop D of the 5′-cloverleaf (5′-CL) of the 5′-NCR initiates the replication of enterovirus RNA [55]. Moreover, the 3D polymerase protein attaches to the viral RNA to form 3D/RNA elongation complexes [56]. Therefore, further studies are needed to determine the relationship between the identified residues on the non-structural proteins and the adaption of CV-A10 in Vero cells.

We compared a variety of CV-A10 strains with different cell adaptation characteristics in Vero cells. We constructed and rescued various recombinant CV-A10 strains to study their cell adaptation properties. Our study highlights the crucial role of CV-A10 structural proteins in cell adaptation, and identifies seven specific amino acid residue mutations that can cooperatively reverse entry ability and improve cell adaptation in Vero cells. Further investigation into recombinant single-site mutated r0195 strains revealed the importance of V1236 in vRNA release during viral uncoating, which ultimately determines CV-A10’s adaptation to Vero cells. These findings provide a foundation for screening potential CV-A10 vaccine candidate strains.

## Figures and Tables

**Figure 1 viruses-15-02114-f001:**
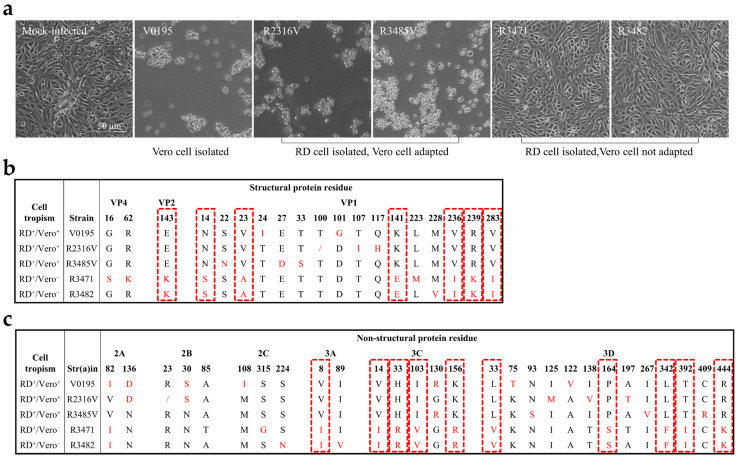
Comparison of CV-A10 strains with different cell tropism. (**a**) The presence and absence of the cytopathic effect (CPE) of different CV-A10 strains in Vero cells were observed at 2 days post-infection using mock-infected cells as a control. The genomes of CV-A10 strains were amplified and sequenced, and the amino acid sequences were aligned. (**b**,**c**) The differential amino acids in the structure proteins and non-structural proteins in different CV-A10 strains are indicated in red. The uniform variant residues related to the same or different cell tropism of the CV-A10 strains are highlighted by red dotted rectangles.

**Figure 2 viruses-15-02114-f002:**
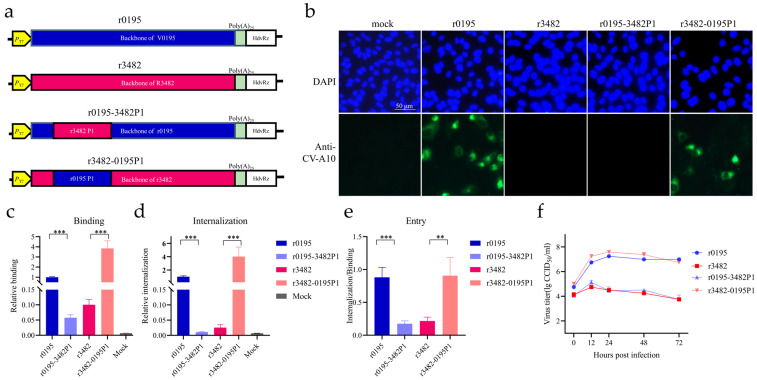
Mapping of the determinants related to growth and cell entry using recombinants. (**a**) Strategy for the construction of recombinants with P1 replacement based on parental strains. (**b**) Vero cells were infected with recombinants as indicated, and indirect immunofluorescence (IFA) was conducted. The structural proteins of CV-A10 (green) and nucleus (blue) were immunostained with rabbit anti-CV-A10 virion antiserum and stained with DAPI, respectively. (**c**,**d**) The binding and internalization of viruses to Vero cells were quantified by RT-qPCR. The content of genomic RNA of binding/internalized viruses was calculated relative to the level of β-actin using the formula 2^[Ct(β-actin)−Ct(vRNA)]^. The relative fold change of the r0195 strain was calculated using delta delta Ct. (**e**) The entry ability was expressed as the ratio of internalized virus relative to the bound virus. (**f**) The viral proliferation ability in Vero cells was displayed using one-step growth curves. The data represent the means ± SDs of values from at least three independent biological samples (n = 3–4). **, *p* < 0.01; ***, *p* < 0.001.

**Figure 3 viruses-15-02114-f003:**
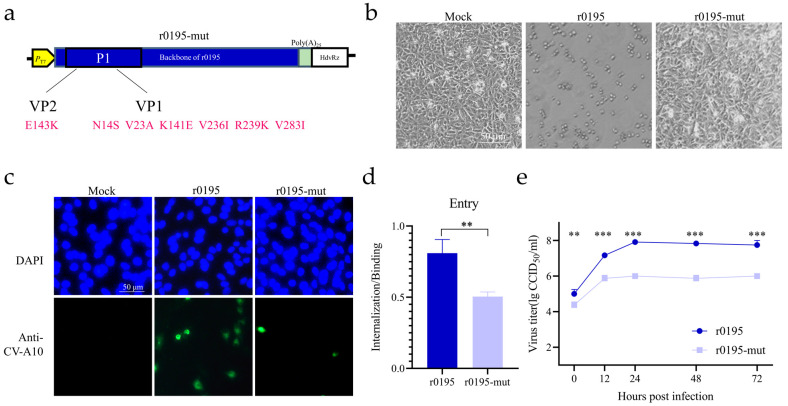
Effect of seven mutations of the structural proteins on cell tropism. (**a**) Construction of the r0195-mut with seven amino acid mutations of r3482 introduced into the structural proteins of r0195. (**b**) The CPE of mock-, r0195- and r0195-mut-infected Vero cells were compared. (**c**) Vero cells were infected with r0195 and r0195-mut. The structural proteins of CV-A10 (green) and the nucleus (blue) were immunostained with rabbit anti-CV-A10 virion antiserum and stained with DAPI, respectively. (**d**) The entry ability was expressed as the ratio of internalized viruses relative to the bound viruses. (**e**) The viral proliferation ability in Vero cells was displayed with one-step growth curves. The virus titer was quantified via a CCID_50_ assay at the indicated time points. The data represent the means ± SDs of values from at least three independent biological samples (n = 3–4). **, *p* < 0.01; ***, *p* < 0.001.

**Figure 4 viruses-15-02114-f004:**
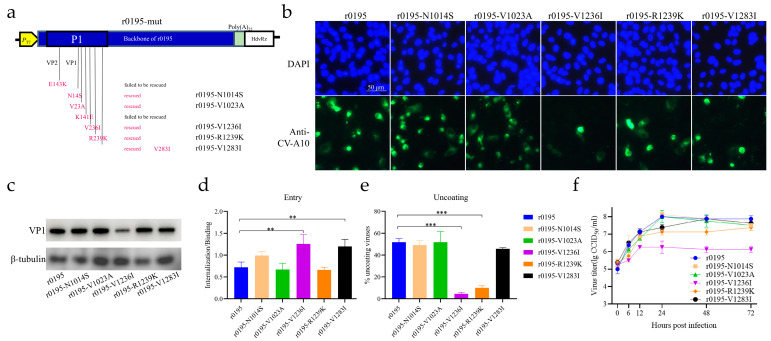
Effect of single-point mutation of the structural proteins of r0195 on cell adaption. (**a**) Construction of mutants with a single amino acid mutation on the structural proteins of r0195. The mutations were E2143K (E to K mutation at amino acid residue 143 of VP2), N1014S (N to S mutation at amino acid residue 14 of VP1), V1023A, K1141E, V1236I, R1239K, and V1283I. Recombinants r0195-E2143K and r0195-K1141E failed to be rescued. (**b**) Vero cells were infected with the five rescued rCV-A10 mutants and the parental r0195. The structural proteins of CV-A10 (green) and the nucleus (blue) were immunostained with rabbit anti-CV-A10 virion antiserum and stained with DAPI, respectively. (**c**) The expression the of VP1 in Vero cells infected with r0195 and the mutants was detected by Western blotting. β-tubulin was used as the loading control. (**d**) The virus entry ability was expressed as the ratio of internalized viruses relative to the bound viruses. (**e**) The viral uncoating ability was evaluated via a plaque assay using light-sensitive NR-CV-A10 recombinant viruses. The viral uncoating efficiency was calculated by dividing light-exposed PFU by dark-kept PFU. (**f**) The viral proliferation ability in Vero cells was displayed with one-step growth curves. The data represent the means ± SDs of values from at least three independent biological samples (n = 3–4). **, *p* < 0.01; ***, *p* < 0.001.

**Figure 5 viruses-15-02114-f005:**
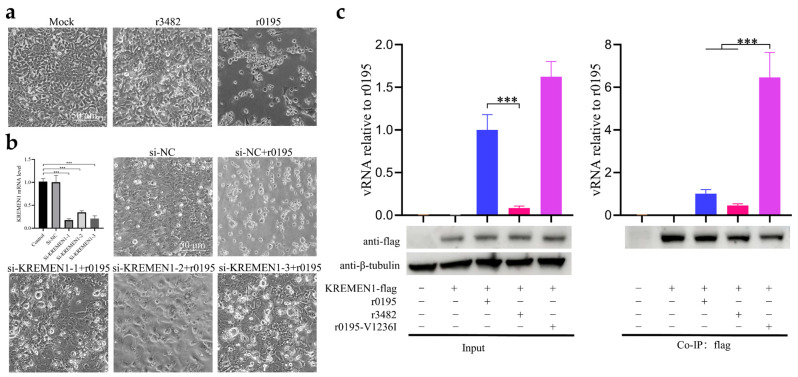
KREMEN1-dependent viral infection and RNA release of KREMEN1-captured virions. (**a**) Normal HEK293 cells were infected with r3482 and r0195 at a m.o.i of 1 or mock-infected, and CPE was observed at 2 days post-infection. (**b**) The interference efficiency of three siRNA was evaluated by qPCR. After siRNA interference, the HEK293 cells were inoculated with r0195 and the CPE was observed at 2 days post-infection. (**c**) HEK293 cells were mock-transfected or transfected with pcDNA3.1-KREMEN1-flag followed by infection with r0195, r3482, r0195-V1236I or mock-infection. Co-immunoprecipitation (Co-IP) was conducted, and the over-expressed KREMEN1 in cell lysis and the immunoprecipitated KREMEN1 were detected using rabbit anti-flag antibody. β-tubulin was used as a loading control in Western blotting. The viral RNA of virions entered into the cells, and the unreleased viral RNA in the KREMEN1-co-immunoprecipitated virions was quantified via RT-qPCR. The data represent the means ± SDs of values from at least three independent biological samples (n = 3–4). ***, *p* < 0.001.

## Data Availability

Not applicable.

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
