# Peer review of "Identification of Critical Amino Acids of Coxsackievirus A10 Associated with Cell Tropism and Viral RNA Release during Uncoating"

_viruses, 2023, doi:10.3390/v15102114_

Round 1
Reviewer 1 Report
In manuscript written by Jie Pei , Rui-Lun Liu , Zhi-Hui Yang , Ya-Xin Du , Sha-Sha Qian , Sheng-Li Meng , Jing Guo , Bo Zhang and Shuo Shen authors compared a variety of CV-A10 strains with different cell adaptation characteristics in Vero cells. They constructed and rescued various recombinant CV-A10 strains to 497 study their cell adaptation properties. Present study highlights the crucial role of CV-A10 498 structural proteins in cell adaptation and identifies seven specific amino acid residue mutations that can cooperatively reverse entry ability and improve cell adaptation in Vero cells. Importance of these studies include investigation into the recombinant single-site mutated r0195 strains. Results revealed the importance of V1236 in vRNA release during viral uncoating, which ultimately determines CV-A10’s adaptation to Vero cells. These findings provide a foundation for screening potential CV-A10 vaccine candidate strains.
This manuscript is written well, results are well presented in manuscript and conclusions are based on obtained results. Overall this is a manuscript with combinations of different virological methods to describe molecular virology adaptations of Coxsackevirus A10 on permissive and nonIn manuscript written by Jie Pei , Rui-Lun Liu , Zhi-Hui Yang , Ya-Xin Du , Sha-Sha Qian , Sheng-Li Meng , Jing Guo , Bo Zhang and Shuo Shen authors compared a variety of CV-A10 strains with different cell adaptation characteristics in Vero cells. They constructed and rescued various recombinant CV-A10 strains to 497 study their cell adaptation properties. Present study highlights the crucial role of CV-A10 498 structural proteins in cell adaptation and identifies seven specific amino acid residue mutations that can cooperatively reverse entry ability and improve cell adaptation in Vero cells. Importance of these studies include investigation into the recombinant single-site mutated r0195 strains. Results revealed the importance of V1236 in vRNA release during viral uncoating, which ultimately determines CV-A10’s adaptation to Vero cells. These findings provide a foundation for screening potential CV-A10 vaccine candidate strains.
This manuscript is written well, results are well presented in manuscript and conclusions are based on obtained results. Overall this is a manuscript with combinations of different virological methods to describe molecular virology mechanisms of adaptations of Coxsackevirus A10 on permissive and non permissive cell lines. I have two suggestions:
Figure 4.b Images with only DAPY I can see a slight green staining, It will be better if the authors will clear up DAPI images from green contaminated staining and indicate the size of cells.
I am just curious about stability of described mutations on Vero cells propagations. I would like to know how many passages on Vero cells described adapted strains of CV-10 get before sequence.
Reviewer 2 Report
The authors aimed to explore the variation in amino acid sequence which allows or prevents efficient replication of coxsackievirus A10 (CVA10) in Vero cells given the advantage of Vero cells for generating vaccine preparations. Using recent CVA10 isolates which were isolated from RD cells but were passaged to adapt to Vero cells, strains which did not adapt to Vero cells in passage and one Vero isolate which could replicate in RD as well, sequences were obtained to identify variations correlated with the ability to replicate in Vero. Amino acid sequence analysis identified 6 sites in the capsid region and 10 in the nonstructural coding regions which varied according to whether the virus strain would replicate only in RD or in RD and Vero cells. The authors cloned one of each strain, r0195 (replicates in both cell cultures) and r3482 (replicates only in RD). Using these clones, the authors generated recombinant genomes with the P1 of r3482 in the r0195 background (r0195-3482P1) and the reverse recombinant r3482-0195P1. Generating virus from these strains demonstrated the inability of the r3482 strain and the r0195-3482P1 to express viral protein in Vero cells compared to the r1095 and r3482-0195P1 as well as demonstrating that the binding and internalization of virus in Vero cells correlated with the 0195 P1, thus demonstrating the importance of the variant sites in the P1 region. Using mutagenesis of the 6 sites in the r0195 background to the 3482 variation at those sites followed by generation of virus with single mutations at those sites in the r0195 background, the authors found that only 4 of the 6 mutants could be rescued from RD cell expression, and only two had significant effects upon replication in Vero cell culture, affecting virus uncoating rather than binding and entry. Only the V1236I mutation in P1 decreased the replication in Vero cells significantly. The authors provided an interesting proof of the uncoating effect by silencing the CVA10 receptor, KREMEN1 in HEK293 cells and overexpressing a KREMEN1-Flag protein. CVA10 virus coprecipitating with the KREMEN1-flag had 6 times the amount of viral genome present when the infecting virus was the V1236I mutant rather than the r0195 parental strain.
It is interesting that 1236I is a less common variant in the GenBank CVA10 sequences and it would be expected that the use of RD cells for isolation of enterovirus As would prevent a selection against 1236I if this variation was critical alone for the inability to replicate in Vero cells. The authors cited the inability to isolate CVA10 isolates capable of growing in Vero cells without adaption to this cell line. I know that the array of GenBank CVA10 sequences contains older strains and the isolation of the CVA10 from cultures varies. Are there any studies of CVA10 in which culture was not performed but sequencing from the human samples was done directly to address this question?
A more significant issue that I had with these studies was whether the genomes of the viruses generated by RD culture corresponded to the cloned genomes? Also, when the authors passed these recombinant or mutagenized viruses in Vero cells, did the population of genomes from that passage, largely retain the Vero resistant variant or was there a partial reversion at this one site or a compensatory mutation in the P1 region. If the authors did not sequence the population from the cloned genomes expression in RD cells, this must be done to verify the experimental basis of the finding of the importance of 1236 V in Vero adaption. However, if they found passage in Vero caused reversion of this one site or generation of a compensatory mutation, there would be greater significance to the work and possibly a structural understanding to the effect.
Minor issues:
On lines 41-42: I’m not sure what conformational dependent means. Do the authors mean the neutralizing epitopes are largely complex or composite (containing residues from multiple capsid proteins)? Or do they mean that the neutralizing epitope has multiple conformations? Reference 11 does state that a couple of neutralizing antibodies can induce conformational changes in some enterovirus types. Another reference which is much more extensive (Huang KA. Structural basis for neutralization of enterovirus. Curr Opin Virol. 2021 Dec;51:199-206. doi: 10.1016/j.coviro.2021.10.006) reviews most of the HFMD viruses and notes that structural analysis of neutralization indicates interference with virus: receptor binding or induction of conformational changes have been noted for them. This would be a better reference, but Huang et al. does not indicate that the majority of dominant epitopes binding neutralizing antibodies neutralize virus by inducing conformational changes.
Lines 74-77: Is this reference (27) which refers to the role of the 5’NTR, the best reference for the P2 and P3 structural proteins?
Line 98-100: “The coxsackievirus A10 strain CV-A10-V0195/PX 98 CHN/2019 (abbreviated as V0195) was isolated directly in Vero cells from a clinical sample of a patient with HFMD in Peixian, Jiangsu, China in 2019.” This strain, so important in this work, should be referenced if published. I note that the authors have provided a GenBank accession number for this strain, which is adequate if the 2019 work was not published.
Lines 123-125: I understand that the entire P1 of these clones was substituted for the sequence of the parental strain. Please provide the nt numbers of the GenBank entries to define the region used.
Lines 164-168: Please reference primers or indicate the corresponding nucleotides in a GenBank entry.
Lines 193-195: Please provide the siRNA sequences or a reference for them.
Lines 202-203 and 392-395 and 453-456: “The plasmid pcDNA3.1-KREMEN1-flag expressing the KREMEN1 receptor was constructed.” And “To investigate whether the key role of V1236 in uncoating is related to the cell receptor KREMEN1, the KREMEN1 gene was cloned from Vero cells and the plasmid pcDNA3.1-KREMEN1-flag expressing the KREMEN1 with a C-terminal flag tag was constructed.” And “The KREMEN1 used in our study was cloned from Vero cells and the amino-acid sequence was aligned with the human KREMEN1 (data not shown). The amino-acid sequence of the KR, WSC and CUB domains of KREMEN1 is highly conserved in monkeys and humans.”
Please provide a reference to the sequences of the human and monkey KREMEN1 or the GenBank accession numbers for the KREMEN1 used for this work.
Line 89: The plural of medium is media. “All media were supplemented…”
Line 127: eukaryocyte should be eukaryotic cell.
Round 2
Reviewer 2 Report
The authors have addressed the issues I raised to my satisfaction.